# Characterization of the Hydrolysis Kinetics of Fucosylated Glycosaminoglycan in Mild Acid and Structures of the Resulting Oligosaccharides

**DOI:** 10.3390/md18060286

**Published:** 2020-05-29

**Authors:** Xixi Liu, Zhexian Zhang, Hui Mao, Pin Wang, Zhichuang Zuo, Li Gao, Xiang Shi, Ronghua Yin, Na Gao, Jinhua Zhao

**Affiliations:** 1School of Pharmaceutical Sciences, South-Central University for Nationalities, Wuhan 430074, China; Lxx201709@163.com (X.L.); Nataliezz@163.com (Z.Z.); wangpin1994@163.com (P.W.); 18271682301@163.com (Z.Z.); glp1284813702@163.com (L.G.); xiangshi041@gmail.com (X.S.); 2State Key Laboratory of Phytochemistry and Plant Resources in West China, Kunming Institute of Botany, Chinese Academy of Sciences, Kunming 650201, China; maohui@mail.kib.ac.cn (H.M.); yinronghua@mail.kib.ac.cn (R.Y.)

**Keywords:** fucosylated glycosaminoglycan, mild acid hydrolysis, hydrolysis kinetics, sulfated fucose, oligosaccharide

## Abstract

Mild acid hydrolysis is a common method for the structure analysis of fucosylated glycosaminoglycan (FG). In this work, the effects of acid hydrolysis on the structure of FG from *S. variegatus* (SvFG) and the reaction characteristic were systemically studied. The degree of defucosylation (DF) and molecular weights (Mw) of partial fucosylated glycosaminoglycans (pFs) were monitored by ^1^H NMR and size-exclusion chromatography, respectively. The kinetic plots of DF, degree of desulfation (DS) from fucose branches, and degree of hydrolysis (DH) of the backbone are exponentially increased with time, indicating that acid hydrolysis of SvFG followed a first-order kinetics. The kinetic rate constants *k_D_*_F_, *k_D_*_S_, and *k_D_*_H_ were determined to be 0.0223 h^-1^, 0.0041 h^-1^, and 0.0005 h^-1^, respectively. The structure of the released sulfated fucose branches (FucS) from SvFG and HfFG (FG from *H. fuscopunctata*) was characterized by 1D/2D NMR spectroscopy, suggesting the presence of six types of fucose: α/β Fuc2S4S, Fuc3S4S, Fuc3S, Fuc4S, Fuc2S, and Fuc. The Fuc3S4S was more susceptible to acid than Fuc2S4S, and that the sulfate ester in position of O-2 and O-3 than in O-4 of fucose. The structure characteristic of pF18 indicated the cleavage of backbone glycosidic bonds. The APTT prolonged activity reduced with the decrease of the DF and Mw of the pFs, and became insignificant when its DF was 87% with Mw of 3.5 kDa.

## 1. Introduction

Fucosylated glycosaminoglycan (FG) is a unique glycosaminoglycan (GAG) derivative isolated from the body wall of sea cucumbers. FG is typically composed of a chondroitin sulfate-like backbone which consists of *N*-acetyl-d-galactosamine (GalNAc) and d-glucuronic acid (GlcA), connected by alternating β(1→3) and β(1→4) glycosidic bonds, and α-l-sulfated fucosyl branches with distinct sulfated patterns that linked to the position 3 of the GlcA via α(1→3) linkages [1,2,3]. As a heterogenous macromolecule, it has been challenging to obtain the precise structure of native FG. Since the 1980s, the FG structure has been studied using the mild acid hydrolysis, methylation analysis, and 1D/2D NMR spectroscopy. Recently, various depolymerization methods with glycosidic bonds selectivity, such as β-eliminative and deaminative depolymerization, were established to prepare several series of oligosaccharides, which enabled further investigation of the FG complex structure [4,5,6]. 

Mild acid hydrolysis is a common method for the degradation of sulfated fucose (FucS) branches from FG. The glycosidic bonds between FucS branches and the backbone are more susceptible to mild acid hydrolysis than those of GlcA and GalNAc in the backbone [3,7]. The structure of the acid-released fucose branches can be studied via 1D/2D NMR spectroscopy, and the composition of the acid-resistant backbone can be further characterized after chondroitinase degradation. In general, during the acid hydrolysis process, the glycosidic oxygen atom becomes the conjugate acid, which then results in a nonreducing end group by unimolecular heterolysis, and a carbonium-oxonium ion which forms reducing end group quickly [8].

The mild acid hydrolysis method was firstly used in 1987 for the structure study of FG from *Ludwigothurea grisea* (LgFG) by Mourão et al [9]. LgFG contained glucuronic acid and amino sugars in approximately equimolar proportions, and the characterization of sulfated fucose indicated the LgFG was a fucose-rich sulfated polysaccharide [1]. The native FG was totally resistant to chondroitinase degradation, whereas defucosylated FG obtained by mild acid hydrolysis can be degraded by chondroitinase AC. Combined with the methylation and NMR analysis, it was suggested that native FG contained mono- and disaccharide sulfated fucose branches, and that the GlcA was also substituted with sulfate esters at position *O-*3 [10]. However, in 1996, further analyzing the structures of the mild acid hydrolysis products indicated that LgFG contained both mono- and disulfated fucoses as branches, which linked to the *O-*3 position of GlcA in the backbone [3]. The detailed structure of LgFG was further reported by the same researchers in 2007 [11]. It was showed the preponderant types of the GalNAc residues in LgFG were 6-sulfated, 4-sulfated, and 4, 6-disulfated GalNAc with the proportions of 53%, 12%, and 4%, respectively, and the 2,4-disulfated fucose branches linked to the 3-position of the GlcA residues. Considering the desulfation during the mild acid hydrolysis, the intact LgFG was analyzed by NMR directly, and the FucS was confirmed as three sulfated patterns of the side chain, L-Fuc-(α1-2)-L-Fuc3S, L-Fuc2S4S, and L-Fuc3S4S [2]. As mentioned above, even the structure of FG from the same species may be reported inconsistently, partly due to the effect of acid hydrolysis reaction on the structure, which has not been clearly elucidated. For instance, sulfation types in the backbone and branches might vary due to the desulfation in a different degree.

In addition, the mild acid hydrolysis method, followed by chondroitinases and sulfatases degradation, was used for structure analysis of FG from *Stichopus japonicas* (AjFG), and the backbone of AjFG included four types of chondroitin sulfate (chondroitin, CS-A, CS-C, CS-E) [7]. The GalNAc in AjFG was sulfated at both C-4 and C-6 positions, as confirmed by NMR spectroscopy [12]. Except LgFG and AjFG, many other reported FGs were studied using mild acid hydrolysis [2,13,14,15] to give various structures, especially the sulfated types of FucS and GalNAc residues. The mild acid hydrolysis method could effectively demonstrate the general structural differences among FGs from different species, but could not accurately evaluate the structure sequence, due to its extensive desulfation. Additionally, the correlation between the hydrolysis condition and the hydrolysis degree of FG has not been clarified, especially the hydrolysis of sulfate ester and backbone glycosidic bonds, it was essential to study the kinetics process of FG hydrolysis in mild acid.

FG exhibits various biological activities, such as antiviral, antitumor, anticoagulant, and antithrombotic activities [11,16,17]. In particular, FG could be a potent anticoagulant and antithrombotic agent through multiple mechanisms, such as heparin cofactor II-(HCII) dependent inhibition of thrombin (factor IIa) [18], AT-III dependent inhibition of thrombin [19,20], and the potent inhibition of factor Xa generation by the intrinsic tenase complex [21]. The anticoagulant activity of FG depends on the degree of fucosylated and molecular weight, as well as the sulfated pattern of FucS branches [3,4,5,6,7,8]. Those studies on the effect of FucS on the bioactivity of FG were usually studied using the partial fucosylated glycosaminoglycans products obtained by partial acid hydrolysis. However, the potential destruction like desulfation of the acid hydrolysis products has not been considered, which hindered the clarification of the structure-activity relationship. 

In this study, the acid hydrolysis kinetics of a FG from *S. variegatus* (SvFG) was investigated by constantly measuring the variation of molecular weight and the degree of defucosylation (DF) of the hydrolyzed products. The kinetic plots of DF, degree of desulfation in fucose branches (DS), and degree of hydrolysis of glycosidic bonds in the backbone (DH) versus time were analyzed to give the kinetic equation and kinetic parameters. The IR spectrometer, polarimeter, conductimetric, and NMR methods were employed to analyze the structure characteristic of the hydrolyzed derivatives. The released fucose mixtures from SvFG and HfFG (*H. fuscopunctata*) were characterized via 1D/2D NMR spectrometry, and the desulfation in the released fucose was analyzed by comparison of its composition change during hydrolysis. The structure of the partial fucosylated glycosaminoglycans (pFs) was studied by physicochemical property determination and 1D/2D NMR spectroscopy analysis. The effect of Mw, desulfation and defucosylation on anticoagulant activity of pFs was evaluated by the APTT prolonged activity. Based on these analyses, this study may provide valuable data for further application of the acid hydrolysis method in FG structure analysis.

## 2. Results and Discussion

### 2.1. Structures of SvFG and HfFG

The crude polysaccharides were extracted and isolated from the body wall of sea cucumbers *S. variegatus* and *H. fuscopunctata* after papain enzymolysis and alkaline treatment [22,23], respectively. The homogenous SvFG and HfFG were purified by strong anion-exchange chromatography and the yield was 1.1% and 0.8%, respectively. The ^1^H NMR spectra of SvFG and HfFG were shown in Figure 1. The signals in the region of ~1.2–1.9 ppm could be readily assigned to the methyl protons of the FucS and GalNAc, respectively [2]. According to the chemical shifts in the region of 5.17–5.60 ppm, which could be assigned to α anomeric protons of FucS, SvFG and HfFG mainly contained single sulfated type of FucS: Fuc2S4S (type I, 5.60 ppm) and Fuc3S4S (type III, 5.26 ppm), respectively. Their precise structures have been confirmed by a series of pure oligosaccharides from chemoselective depolymerization [5]. SvFG and HfFG were highly regular glycosaminoglycans with definite FucS branches, and they were excellent model compounds for clarifying the effect of mild acid hydrolysis on the structures of FucS branches and the backbone, although they may have a small amount of other sulfated types of FucS (Figure 1C). 

### 2.2. Mild Acid Hydrolysis of SvFG and HfFG

To study the effect of mild acid hydrolysis on the FG structure, the hydrolysis products of SvFG, including the partial fucosylated products (pFs) and the acid-released side chains (Scs), were purified by size-exclusion chromatography (SEC). The pF1-12 and Sc1-12 were the products of SvFG degraded in 0.1 M H_2_SO_4_ at 60 °C for different time (≤36 h); the pF13-16 and Sc13-16 were the hydrolysis products of SvFG at 50, 60, 70, and 80 °C for 5 h, respectively; the pF17–18 was the product of SvFG at 100 °C for 0.5 and 2 h, respectively (Table 1).

### 2.3. Analysis of Defucosylation of the Partial Fucosylated Products by ^1^H NMR 

To analyze the hydrolysis of FucS branches, pFs were analyzed by ^1^H NMR spectroscopy method. Comparison of the ^1^H NMR spectra of SvFG (Figure 1A) and the pF1-12 (Figure 2A), the fucose residues were clearly removed by mild acid hydrolysis. The signals at ~5.6 ppm were the anomeric protons of α-Fuc2S4S, whose integrals decreased gradually with time course (Figure 2B). Simultaneously, the integrals ratio of new signals at 3.4~3.1 ppm assigned to H-2 and H-3 of the unsubstituted GlcA increased concomitantly with defucosylation. The integral ratios of signals at ~2.01 ppm and ~1.2 ppm assigned to protons in the acetyl (3H) of GalNAc and methyl groups in FucS (3H) [5], respectively, were calculated according to their ^1^H NMR spectra to give degree of defucosylation (DF) of pFs. In the time course of the reaction at 60 °C, the DF of pFs clearly increased gradually from 8% to 63% (Table 1). The DF of pF13-16 obtained at a different temperature was 17% ~ 63%. The DF of pF17-18 obtained at 100 °C was 81% and 87%, respectively. 

### 2.4. Physicochemical Properties and Structures of Hydrolysis Products

Physicochemical properties of pF1–18 were summarized in Table 1. According to the calibration curve, the average molecular weight (Mw) of pF1-12 was calculated as 52.45~12.54 kDa (Table 1 and Appendix A), and the Mw-time curve showed that the Mw of pFs dropped gradually as the hydrolysis proceeded (Figure 3A). The Mw of pF13-16 was 47.72~12.91 kDa, indicating the Mw decreased with increasing temperature (Figure 3B). In addition, the Mw of pF17-18 hydrolyzed at 100 ^o^C was 5.33 and 3.50 kDa, respectively. 

The sulfate group content of pF1–18 was measured using a conductimetric method [23] (Table 1). The SO_3_^−^ content of pF1–12 decreased generally as the degradation of sulfated fucose in SvFG during mild acid hydrolysis at 60 °C (Appendix A). Besides, after hydrolysis for 6 h, the SO_3_^−^ content was basically stable at around 22%~26%, which suggests the acid-resistant sulfate group may exist in FG. Additionally, it was supported by pF13–18 with SO_3_^−^ at around 20%~26% obtained at different acid hydrolysis conditions. 

Their IR spectra (Appendix A) showed absorptions at 3454 and 1030 cm^−1^, which could be assigned to the stretching vibration of O–H and C–O, respectively. Absorptions at 1244 and 852 cm^−1^ were derived from the stretching vibrations of S=O and the bending vibration of C-O-S of sulfate ester. The stretching vibration of C=O in GalNAc and GlcA was detected at 1625 cm^-1^, while the symmetric stretch vibration of COO^-^ in GlcA was at 1402 cm^−1^. Additionally, the peak at 2949 cm^-1^ was from C-H stretching vibrations of CH_3_ in Fuc.

In particular, the ^1^H and ^13^C NMR spectra of pF18 with Mw 3.50 kDa was measured to analyze the structure changes (Figure 4). The anomeric signals of Fuc2S4S were observed at 5.4~5.6 ppm and its methyl signals were at 1.1~1.3 ppm. Furthermore, the DF was 87%, calculated by the integral ratio of methyl signals from pF18 and SvFG. After hydrolysis, the chemical shift at 4.90 and 4.75 ppm was assigned to the H-5 and H-4 in FucS, respectively. The protons at 4.40 and 4.44 ppm were the anomeric protons of the internal β-GlcA and β-GalNAc residues, which were similar with the native SvFG. The signals at around 5.1 ppm should be the anomeric protons of α-GalNAc and α-GlcA at the reducing end and the signals at around 4.5 ppm were the corresponding β-configuration anomeric protons, which were confirmed by the cleavage of glycosidic bonds between GlcA and GalNAc in the backbone. The integration of the total anomeric protons of the reducing ends was about 0.30, and the integration of the internal residues including β-GlcA and β-GalNAc residues in the backbone was about 0.62, respectively.

In ^13^C NMR spectra (Figure 4B), the C-6 of FucS at 18.71 ppm was obviously weak due to the extensive defucosylation in FG. The four signals at 94.25/97.99 ppm and 94.79/98.93 ppm were assigned to the C-1 of α-/β-GlcA and GalNAc in reducing end, respectively. The anomeric carbon signal of GalNAc and GlcA in the backbone was detected at 104.27 ppm and 107.20 ppm, respectively. Several signals around 25.04 ppm could be assigned to C-8 of GalNAc. The signal at 64.06 ppm was assigned to C-6 of nonsulfated GalNAc, and the C-6 of the 6-sulfated GalNAc was downshifted to 70.59 ppm. The GalNAc residues were both sulfated at O-4 and O-6 positions in native SvFG, and the presence of the nonsulfated GalNAc in pF18, indicating the desulfation in the backbone during hydrolysis.

### 2.5. Chemical Structures and Compositions of Scs

The acid-released sulfated fucose under different reaction conditions were desalted using a Bio-Gel P-2 column to give Sc3–Sc16. In particular, Sc8 was analyzed by 1D and 2D NMR spectroscopy (Figure 5). The chemical shifts were summarized in Table 2 based on the interpretations of ^1^H-^1^H TOCSY, COSY, and ^1^H-^13^C HSQC spectra (Figure 5C and Appendix A). 

In the ^1^H NMR spectrum (Figure 5A), five signals between 5.2 and 5.6 ppm could be assigned to five α anomeric protons of fucose with distinct sulfation patterns [3,24], since their ^3^*J_H1-H2_* values (3.6–4.6 Hz) were in agreement with that of α-anomers. They could be identified as α anomers of 2,4-*O*-disulfated fucose (Fuc2S4S, type I), 2-*O*-monosulfated fucose (Fuc2S, type II), 3,4-*O*-disulfated fucose (Fuc3S4S, type III), 3-*O*-monosulfated fucose (Fuc3S, type IV), and 4-*O*-monosulfated fucose (Fuc4S, type V), respectively, according to previous reports [3,4,13,15]. The anomeric protons for the β-anomers of Fuc2S4S, Fuc3S4S, Fuc3S, and Fuc4S were between 4.5 and 4.8 ppm. However, it was difficult to assign the β anomeric proton of Fuc4S, for this signal was overlapped with those for H-4 of Fuc2S4S and Fuc4S. The signals at 1.2–1.3 ppm could be readily assigned to the methyl protons of variously sulfated fucose residues and the sulfation pattern would also result in discernible chemical shifts. The corresponding five spin systems could be observed both in ^1^H-^1^H COSY and TOCSY spectra (Figure 5C and Appendix A) and the other protons could be fully assigned (Table 2).

In the ^13^C NMR (Figure 5B) spectrum, five distinct signals were observed from 92 to 99 ppm. In the ^1^H-^13^C HSQC spectrum (Appendix A), the signals at 92–95 ppm were correlated with α anomeric protons of Fuc2S4S, Fuc3S4S, Fuc3S, and Fuc4S, respectively, while the signals at 97–99 ppm correlated with those β anomeric protons. The sulfated positions were confirmed by their C4 between 72 and 84 ppm, which were the C-4 from both α- and β- anomers of Fuc2S4S, Fuc3S4S, Fuc2S, Fuc3S, and Fuc4S, respectively. The signals at around 18 ppm could be readily assigned to the C-6 of those sulfated fucose residues. Total assignment of the carbon signals was achieved (Figure 5B) and showed in Table 2.

The acid-released fragments were the mixture of various sulfated types of fucose. However, its native FG (SvFG) was a highly regular fucosylated glycosaminoglycan and its main fucose branches have been confirmed as Fuc2S4S (85%) using a bottom-up structure analysis strategy [5]. The well-defined structures of pure oligosaccharides derived from deaminative cleavage or the β-eliminative depolymerization method have clarified the precise structure of SvFG. The presence of Fuc2S and Fuc3S in the acid-released mixture indicated the sulfated fucose branches undergo varying degrees of desulfation during the mild acid hydrolysis reaction. 

The fucose mixtures (Sc3-16) obtained from SvFG with different hydrolysis conditions were characterized by ^1^H NMR (Appendix A). According to the chemical shift assignment of various anomeric protons in fucose residues as descried above, the signals between 5.18–5.5 ppm were assigned to six types of α anomeric protons of FucS in Sc3-16, while the signals of corresponding β anomeric protons were 4.55–4.73 ppm (Appendix A). Thus, approximate integrals of anomeric protons and DF were taken into account to calculate the proportion of different sulfated types of fucose residue. 

Comparison of the proportions of various sulfated types of fucose residues in Sc3-12, extensive desulfation were observed in FucS [13,14,15] during the 36 h hydrolysis period at 60 °C (Figure 6A). The content of Fuc2S4S increased from 7.82% to 24.31%, and the Fuc3S4S content changed little (5.45%~6.68%) during this period due to the low content of Fuc3S4S in native SvFG (10%). The amount of Fuc4S increased even more than the original 5% in native SvFG, which indicated that di-sulfated fucoses, especially Fuc2S4S, undergo extensive desulfation. Therefore, the monosulfated fucose (Fuc2S and Fuc3S) should be derived from the desulfation of Fuc2S4S and Fuc3S4S. The proportions of Fuc2S and Fuc3S increased less than Fuc4S (Figure 6A) revealed that 2-*O* and 3-*O* sulfated group in fucose were more liable to acid hydrolysis than the 4-*O*-sulfated group. This conclusion was supported by the acid-released fucose from HfFG. With the desulfation in mild acid hydrolysis, the disulfated fucose could be desulfated to produce monosulfated fucose, and further to produce unsulfated fucose. The presence of unsulfated fucose (0.90%) in Sc10 proved the desulfation. Generally, the hydrolysis degree of sulfate ester in Sc3-12 was increased as the reaction time prolonged (Figure 6B).

The variation of different sulfated fucose types content in Sc13-16, which were obtained by adjusting the temperature, was similar with above (Figure 6C). The Fuc3S4S in SvFG was more susceptible to acid than Fus2S4S by comparing their content ratio at 50 °C, which were 8.70% and 8.06%, respectively. The desulfation degree of Sc13-16 increased from 1.04% to 35.27% as the temperature increased (Figure 6D). 

Unlike SvFG, the FucS branches of HfFG consisted of 5% Fuc2S4S, 85% Fuc3S4S, and 10% Fuc4S [24]. To understand the effect of mild acid hydrolysis on different FG branches, the mixtures of sulfated fucose, Sc19 and Sc20, were obtained from HfFG through mild acid hydrolysis in 0.1M H_2_SO_4_ at 60 °C for 12 h and 100 °C for 2 h, respectively. The ^1^H NMR spectra of Sc19 and Sc20 (Figure 7) showed four identical signals between 5.19–5.27 ppm, which could be identified as α anomeric protons of Fuc3S4S, Fuc3S, Fuc4S, and Fuc (type VI), respectively, due to the ^3^*J*_H1-H2_ value of 3.6–4.2 Hz. Four types of sulfated fucose in Sc19 and Sc20 indicated that the desulfation of FucS during mild acid hydrolysis. The nonsulfated fucose should be from the desulfation of sulfated fucose, and the Fuc2S4S could be negligible basically due to its weak absorption. The approximate integration proportions of the four anomeric signals (type III:IV:V:VI) of 43:4:13:2 in Sc19 and 11: 22: 52: 7 in Sc20 indicated the 3-O-sulfated group was more liable to mild acid hydrolysis than the 4-O-sulfated group. 

Comparison of the ^1^H NMR spectrum of Sc9 and Sc18 (Figure 7), whose hydrolysis condition was the same with Sc19 and Sc20, respectively, the desulfation also presented in released branches from SvFG at different conditions. It was further supported by their composition ratio of diverse fucose (type I:II:III:IV:V:VI) with 19:4:5:2:22:0 in Sc9 and 22:12:3:4:36:10 in Sc18. Besides, we found the sulfate group at 2-O was more liable to acid hydrolysis than 4-O in sulfated fucose.

With the analysis on the structure of the Sc19 and Sc20 mixture, respectively, the chemical shifts of ^1^H and ^13^C (Appendix A) of type III, IV, and V were consistent with the corresponding signals of these monosaccharides in Sc8. The ^1^H and ^13^C chemical shifts of unsulfated fucose are mostly shown in Table 2, however, the carbon signals of α-Fuc were difficult to define due to its trace amount. Comparison of unsulfated fucose, the protons signals in sulfated position showed strong downfield shift, and their adjoining protons signals showed weak downfield shifts [13].

The sulfated fucose in native SvFG and HfFG was the highly regular type with Fuc2S4S and Fuc3S4S, respectively. However, the acid released side chain was a mixture of various sulfated fucose types including α/β Fuc2S4S, Fuc3S4S, Fuc2S, Fuc3S, Fuc4S, and Fuc according to their ^1^H NMR analysis. The presence of monosulfated and nonsulfated fucose indicated the desulfation in the fucose branches during mild acid hydrolysis, and the sulfate group at the O-4 position of fucose residue was more stable than O-2 and O-3 in acid. 

### 2.6. Kinetics Process of SvFG in Mild Acid Hydrolysis 

The hydrolysis of most linear polysaccharides in mild acid followed the first-order kinetics [8,25,26,27]. Additionally, the mild acid hydrolysis of a spruce polysaccharides containing galactose branches was also the first-order kinetics process [25]. In the mild acid hydrolysis process of FG, a small amount of sulfate ester and backbone glycosidic bonds could also be hydrolyzed [3,13], which could affect the FG chemical structure analysis. FG is a linear polysaccharide with large amounts of fucose branches and the hydrolysis process in 0.1 M H_2_SO_4_ at 60 °C for different times was studied. To clarify the hydrolysis kinetics process of FG, the DF of pF1-12, the degree of desulfation in fucose branches containing the acid-released part and those in pFs (DS), and the degree of hydrolysis (DH) of glycosidic bonds in backbone were analyzed. 

After hydrolysis, the DF of pF1-12 increased with the hydrolysis proceeding (Table 1) and the process followed the first-order kinetics (Figure 2B and 8A). The fitting formula was DF = 5.77265 + 0.54881 × e^-2.13151t^ (r^2^ = 0.93). Therefore, the sulfated fucose branches hydrolysis kinetics constant (*k_D_*_F_) was calculated as 0.0223 h^−1^.

The DS increased as the reaction time prolonged (Figure 8B). This process also followed first-order kinetics hydrolysis, and the fitting formula was DS = 0.17593 − 0.13692 × e^-0.04690t^ (r^2^ = 0.94). The kinetic rate constant of DS (*k_D_*_S_) was 0.0041 h^−1^, meaning that the rate of the fucose branches hydrolysis is about 5-fold of that of the sulfate ester hydrolysis. 

The DH calculated using the DF and DS value increased with prolonged time (Figure 8C). The cleavage of the backbone glycosidic bonds followed the first-order hydrolysis basically, and the fitting formula was DH=0.02144-0.02284e^-0.04227t^ (r^2^ = 0.84). The kinetic rate constant of DH (*k_D_*_H_) was 0.0005 h^−1^, less 45 times than *k_D_*_F_, suggesting the α(1→3) glycosidic bonds between fucose and GlcA and the alternating β(1→3) and β(1→4) linkages in the backbone were cleaved during mild acid hydrolysis, and those α(1→3) linkages degraded at a higher rate than β(1→3) and β(1→4) linkages. It was consistent with the preferential cleavage of fucose branches in mild acid hydrolysis. 

Based on the analysis above, the mild acid hydrolysis of SvFG followed the first-order kinetics process basically. The hydrolysis kinetic rate of defucosylation was higher than desulfation in the fucose branches and the cleavage of the glycosidic bonds in the backbone (*k_D_*_F_ > *k_D_*_H_ > *k_D_*_S_). Additionally, the hydrolysis of sulfate ester and backbone glycosidic linkages cleavage obviously increased as the reaction time prolonged, which would affect the analysis of the FG chemical structure, such as the sulfated patterns of fucose and the connection site of branches to the backbone. 

### 2.7. Effects of DF, Mw, and SO_3_^-^ Content on APTT Prolongation Activity in Vitro

The effects of DF, Mw, and SO_3_^-^ content of pFs on their anticoagulant activities about the intrinsic pathways of the coagulation cascade were evaluated using the activated partial thromboplastin time (APTT) assays. The results (Figure 9) showed that the native SvFG showed strong APTT prolonged activity (2.12 μg/mL) as reported [5]. The APTT prolonged activities of the compounds pF10–12 and pF17-18, decreased from 3.97 to > 128 μg/mL as their DF increased from 46% to 87% and decreased with their Mw reduction from 25.6 to 3.5 kDa. In particular, the activity of pF12 (12.5 kDa, 27.23 μg/mL) was about 5.5 times lower than that of pF11 (19.8 kDa, 4.49 μg/mL), whose DF were 52% and 62%, respectively. The compound pF17 with DF of 81% and Mw of 5.3 kDa showed weak activity (41.88 μg/mL), and pF18 with DF 87% and Mw of 3.5 kDa showed negligible activity. These results indicated that the fucose branches and the molecular weight are required for the anticoagulant effect of FG on plasma APTT prolongation. In addition, for pF10–12 and pF17–18 have similar SO_3_^−^ content (20%–26%), the APTT activities might have little correlation with its sulfate content [18]. The APTT prolongation activities of the partial fucosylated glycosaminoglycans were mainly related to the DF and Mw.

## 3. Materials and Methods

### 3.1. Materials

Dried sea cucumbers *S. variegatus* and *H. fuscopunctata* were purchased from local markets in Zhanjiang, Guangdong Province, China. Amberlite FPA98Cl ion exchange resin was purchased from Rohm and Haas Company (Midland, MI, USA). Polyacrylamide gels (Bio-Gel P2, P6 and P10) were from Bio-Rad (Hercules, CA, USA). Sephadex G-25 and G-10 were from GE Healthcare. Deuterium oxide (99.99%) was from Sigma Aldrich (St. Louis, MI, USA).

### 3.2. Isolation of Fucosylated Glycosaminoglycan from the Body Wall of Sea Cucumber

Two kinds of fucosylated glycosaminoglycans (SvFG and HfFG) were extracted from the sea cucumbers *S. variegatus* and *H. fuscopunctata*, respectively, according to previous method [17,28]. The dried powder of the sea cucumber body walls was digested by 0.1% papain in aqueous solution at 50 °C for 6 h and then hydrolyzed by 0.5 M NaOH at 60 °C for 2 h. After removing the protein at pH 2.8, the extracted solution was treated by ethanol and centrifuged at 4700 rpm for 15 min. The crude polysaccharides were purified by strong anion exchange chromatography using Amberlite FPA98 resin eluted with different gradients of NaCl solution. Finally, the FG fractions were obtained after being desalted and freeze-dried. 

### 3.3. Mild Acid Hydrolysis of Fucosylated Glycosaminoglycan

Twelve SvFG (160 mg each) samples were submitted to mild acid hydrolysis [18] in 32.0 mL of 0.1 M H_2_SO_4_ at 60 °C, the reaction time course was 0.25, 0.5, 1, 1.5, 2, 3, 4.5, 6, 12, 18, 24, and 36 h, respectively. Another four SvFG (160 mg each) samples were submitted to mild acid hydrolysis in 32.0 mL of 0.1 M H_2_SO_4_ at 50, 60, 70, and 80 °C for 5 h, respectively. About 500 mg of SvFG was subjected to hydrolyze at 100 °C for 0.5 and 2 h. After neutralization with NaOH, the concentrated hydrolysates were separated by Sephadex G-25 column eluted with distilled water. The partial fucosylated glycosaminoglycan (pF1–18) and a mixture of sulfated fucose (Sc1–18) were individually pooled and lyophilized. 

About 250 mg HfFG each was subjected to partial acid hydrolysis in 50 mL of 0.1 M H_2_SO_4_ at 60 °C for 12 h or at 100 °C for 2 h. The subsequent purification procedures were similar to the above steps, and the partial fucosylated glycosaminoglycan (pF19–20) and the acid-released fucose (Sc19–20) were obtained. 

### 3.4. Purification of the Acid-released Fucose Fragments

The acid-released fucose was desalted by Bio-Gel P-2 column (110 × 0.8 cm) and eluted with distilled water at a flow rate of 6 mL/h. The collected fractions were monitored by the phenol-H_2_SO_4_ method [3,29]. The fractions related to fucose were pooled and lyophilized to give Sc3–20 (Sc1–2 was difficult to collect enough amounts for analysis).

### 3.5. Determination of Physicochemical Properties

The molecular weight (Mw) of pF1-20 was determined by HPGPC using an Agilent Technologies 1260 series (Agilent Technologies, Santa Clara, CA, USA) apparatus equipped with a Shodex OH-pak SB-804 HQ column (8 mm × 300 mm) and differential refractive index (RI) detector. Chromatographic conditions were performed as the previous method [18,28]. The Mw was calculated by GPC software (Version 3.4) using a curve fitted by a serials of FG samples with known Mw (52.77, 39.9, 27.76, 14.92, 8.24, 5.30, 3.12 kDa). 

The sulfate ester content of pF1-20 was measured by a conductimetric method [23]. The pFs (4 mg) was dissolved in 2.0 mL of distilled water. The solution was passed through a column (10 × 200 mm) of Dowex® 50WX8 cation exchange resin (H^+^ form), and the acid form of these samples was eluted with water and then the solution was titrated with 2 mM NaOH at room temperature, monitored by a conductivity meter (DDSJ-308A). The sulfate ester content was calculated from the conductivity titration curves. 

The FI-IR spectra (KBr pellets) of SvFG and pF1–12 (~1 mg) were recorded on a Bruker Tensor 27 infrared spectrometer (Ettlingen, Germany) in the range of 400–4000 cm^−1^. 

### 3.6. NMR Method

The ^1^H NMR spectra of the pFs and Scs were recorded in D_2_O on a Bruker Avance III-600 MHz spectrometer equipped with a ^13^C/^1^H dual probe in FT mode. To analyze the structure characteristics of hydrolysis products, the ^1^H/^13^C, COSY, TOCSY, HMBC, and HMQC spectra of Sc8, Sc19, Sc20, and pF18 were recorded at 298 K using standard Bruker pulse sequences. 

### 3.7. Determination of the Kinetic Parameters 

The degradation of most linear polysaccharides in mild acid hydrolysis followed the first-order kinetics [25,26]. The kinetics process of the mild acid hydrolysis of FG including the degree of defucosylation (DF), the degree of desulfation in side chains (DS), and the degree of hydrolysis of glycosidic bonds in the backbone (DH) [13] were analyzed. The calculation formula of DF was given as:(1)$$\mathrm{DF}\;(\%)=(1-\frac{A_{f}}{A_{a}})\times 100\%,$$
where *A_f_* and *A_a_* denote the methyl integrals of L-Fuc and D-GalNAc in the ^1^H NMR spectra of pFs, respectively. 

The calculation formula of DS was given as:(2)$$\mathrm{DS}\;(\%)=\{[(1-\mathrm{DF}{)}_{\;}\times_{\;}(\sum_{i=0}^{2}x\prime_{i}y\prime_{i}{)}_{\;}+{\mathrm{DF}}_{\;}\times_{\;}(\sum_{i=0}^{2}x{"}_{i}y{"}_{i}){]}_{\;}/1.95{\}}_{\;}\times_{\;}100\%,$$
where 1.95 is the degree of sulfation of FucS in the native SvFG; x’*_i_* and x”*_i_* are the mole percentage of non-, mono-, disulfated fucose in pFs and Scs, respectively; y’*_i_* and y’’*_i_* are the sulfate group number (0, 1, and 2). 

The calculation formula of DH was given as:DH (%) = [(Mw_c_ – Mw_t_) / (n × Mw_t_)] × 100%,(3)
where n is the number of trisaccharide units in native FG (~ 55); Mw_t_ is the average molecular weight of pFs as described in 3.5; Mw_c_ is the molecular weight of pFs’ backbone when only the sulfate group and fucose are hydrolyzed, and its calculation formula was given as:Mw_c_ = n × {200 + [203 + 2 × (1 − DS) × 102] + (1 − DF) × [164 + 1.95 × (1 − DS) × 102]},(4)
where 200, 203, and 164 are the molecular weight of GlcA, GalNAc, and Fuc, respectively; 102 is the sulfate group substitution increased; 2 and 1.95 are the number of sulfate groups in GalNAc and Fuc of native SvFG, respectively; DS and DF are as described above.

The kinetic parameters were estimated by fitting curves using the Equation (5):DF or DS or DH = e^*kt*^.(5)

### 3.8. Determination of the APTT Activities in Vitro

The active partial thromboplastin time (APTT) of SvFG and its partial fucosylated derivatives (pF10–12 and pF17–18) were detected with a coagulometer (TICO MC-2000, Hagen, Germany) using the APTT reagents (MDC Hemostasis, Hagen, Germany). Normal human plasma was re-dissolved in 4 mL distilled water. The test samples were dissolved and diluted by 20 mM Tris-HCl (pH 7.4) at various concentrations. Human plasma (45 μL) and the sample (5 μL) were mixed and incubated for 2 min at 37 °C. Further, the APTT reagent (50 μL) was added to the mixture. After incubation for 3 min at 37 °C, the CaCl_2_ solution (50 μL) was added, and then the clotting time was recorded.

## 4. Conclusions 

The unique FG chiefly consisted of α(1→3) glycosidic linkages between L-FucS and D-GlcA, alternating β(1→3) and β(1→4) linkages between D-GlcA and D-GalNAc. The mild acid hydrolysis of FG based on the α(1→3) glycosidic linkages is more susceptible to acid than those in the backbone widely used in the structure analysis of FG. In our study, a highly regular FG from the sea cucumber *S. variegatus* was used to investigate the mild acid hydrolysis kinetics of FG, including the sulfate group hydrolysis, glycosidic bonds hydrolysis in branches and/or backbone.

The physiochemical properties including the Mw, DF, and sulfate groups content of the partial fucosylated derivatives were determined and compared, indicating the alternating β(1→3) and β(1→4) linkages in the backbone were also cleaved during mild acid hydrolysis and the sulfated groups in the GalNAc were also partial hydrolyzed. Unfortunately, the reducing ends, the non-reducing ends, the location of the residue fucose branches, and the sulfated position in the fucose or the GalNAc residues may be difficult to clarify according to current results. The investigation of the purified oligosaccharides from the pFs is in progress and will be published in due course. 

In addition, the structures of the released sulfated fucoses were elucidated by ^1^H, ^13^C, and 2D NMR, which were composed of various sulfated types including Fuc2S4S, Fuc3S4S, Fuc3S, Fuc4S, and Fuc2S. The sulfated types were more complex than the highly regular sulfated fucose branches in the native SvFG (Fuc2S4S) or HfFG (Fuc3S4S), which have been confirmed according to the precise structure of the purified oligosaccharides from its β-eliminative depolymerized products. Thus, to elucidate the structure of sulfated branches of native FG from the acid-released fucose mixture would be hindered due to its extensive desulfation during hydrolysis. The desulfation results of acid-released fucose from SvFG and HfFG indicated that the Fuc3S4S was more susceptible to acid than Fuc2S4S during mild acid hydrolysis and the sulfate group at the O-4 position of fucose residue was more stable than O-2 and O-3 in acid. 

The hydrolysis kinetics process of defucosylation (DF), desulfation in fucose branches (DS), and glycosidic bonds cleavage in the backbone (DH) in acid followed first order kinetics reaction. The hydrolysis kinetics rate of defucosylation was higher than desulfation in fucose branches and the cleavage of the glycosidic bonds in the backbone (*k_D_*_F_ > *k_D_*_H_ > *k_D_*_S_).

Overall, the cleavage of the backbone glycosidic bonds and the desulfation during hydrolysis could mislead FG structure analysis. Thus, mild acid hydrolysis is not a very suitable method to analyze the precise structure of FG, and the bottom-up strategy based on depolymerization methods, which could selectively cleave glycosidic bonds in the backbone like the β-eliminative or deaminative depolymerization method, could be useful for the structure analysis of native FGs from different species.

In biological activity, the APTT prolonged activity reduced due to the variation of degree of fucosylation and the molecular weight in hydrolysis products. When its DF increased to 87% with Mw of 3.5 kDa, the anticoagulant activity became insignificant.

## Figures and Tables

**Figure 1 marinedrugs-18-00286-f001:**
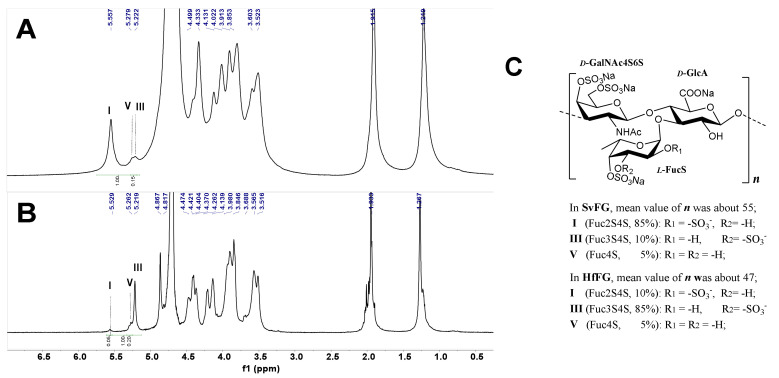
^1^H NMR spectra of SvFG (**A**) and HfFG (**B**) and their structures (**C**). Labels I, III, V represent type I (Fuc2S4S), III (Fuc3S4S), V (Fuc4S) of FucS, respectively.

**Figure 2 marinedrugs-18-00286-f002:**
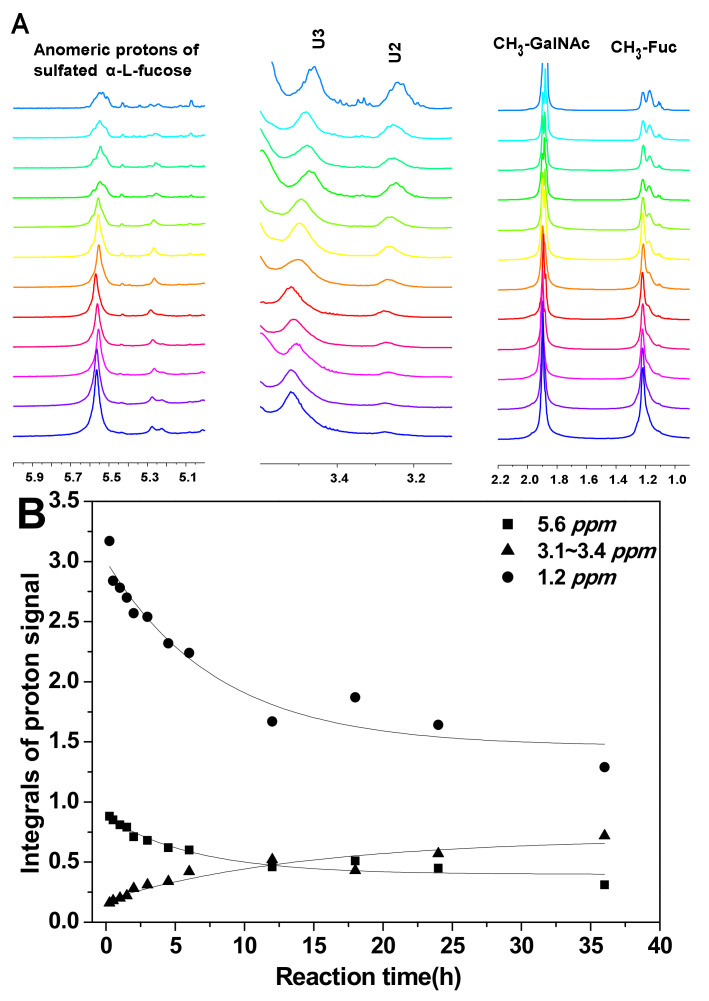
Part of the ^1^H NMR spectra of pFs at different periods of partial acid hydrolysis at 60 °C (**A**); the integral variation of proton signals at 5.6, 3.13, 1.2 ppm as hydrolysis time course (**B**).

**Figure 3 marinedrugs-18-00286-f003:**
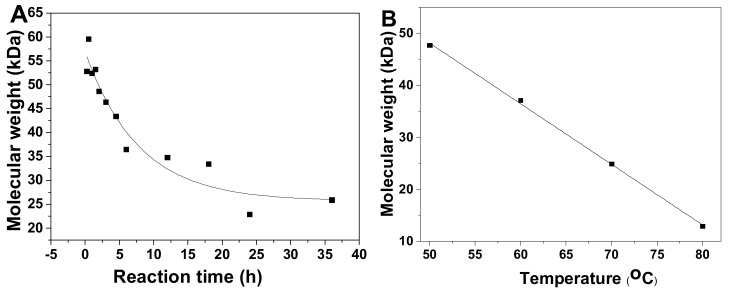
Time course (0–36 h) of average molecular weight (Mw) of pF1–12 (**A**) and temperature course (50, 60, 70 and 80 °C) of Mw of pF13–16 (**B**).

**Figure 4 marinedrugs-18-00286-f004:**
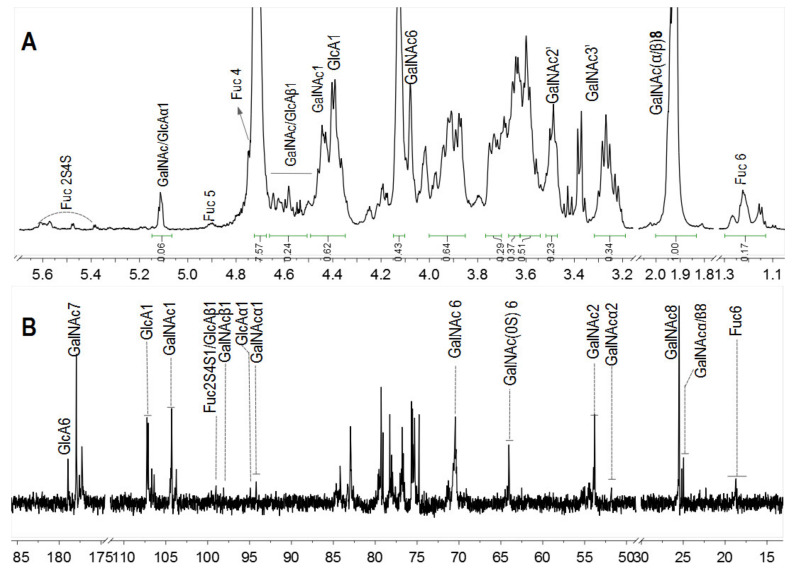
^1^H (**A**) and ^13^C (**B**) NMR spectra of pF18.

**Figure 5 marinedrugs-18-00286-f005:**
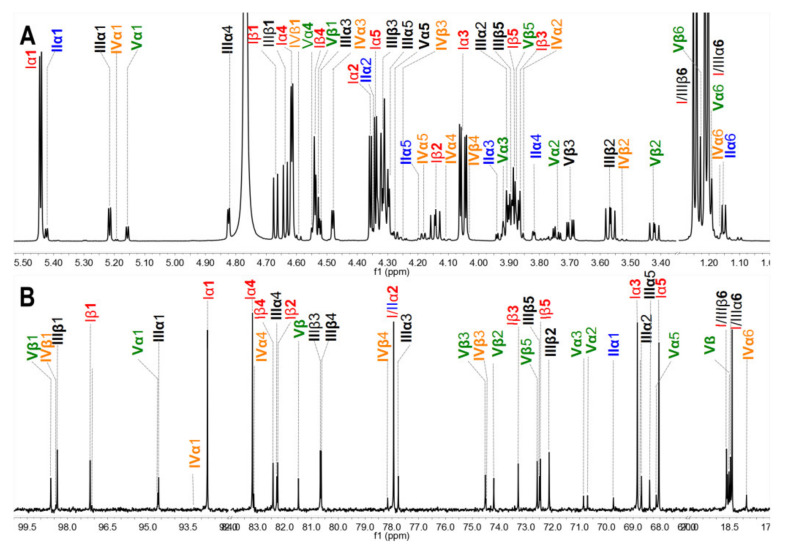
^1^H /^13^C NMR (**A**, **B**), ^1^H-^1^H COSY spectra of Sc8 (**C**) and the structures of Sc8 (**D**).

**Figure 6 marinedrugs-18-00286-f006:**
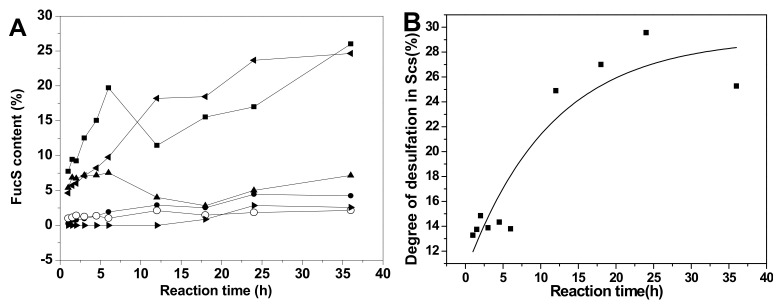
The variation of sulfated patterns of fucose released from SvFG with the reaction time (**A**) and temperature (**C**), the fitting curve of degree of desulfation in Scs versus reaction time (**B**) and temperature (**D**). Percentage proportions of Fuc2S4S (■), Fuc2S (●), Fuc4S (◂), Fuc3S4S (▴), Fuc3S (○), and Fuc (▸) were determined from the ^1^H NMR spectra of the acid-released and corresponding acid-resistant fragment.

**Figure 7 marinedrugs-18-00286-f007:**
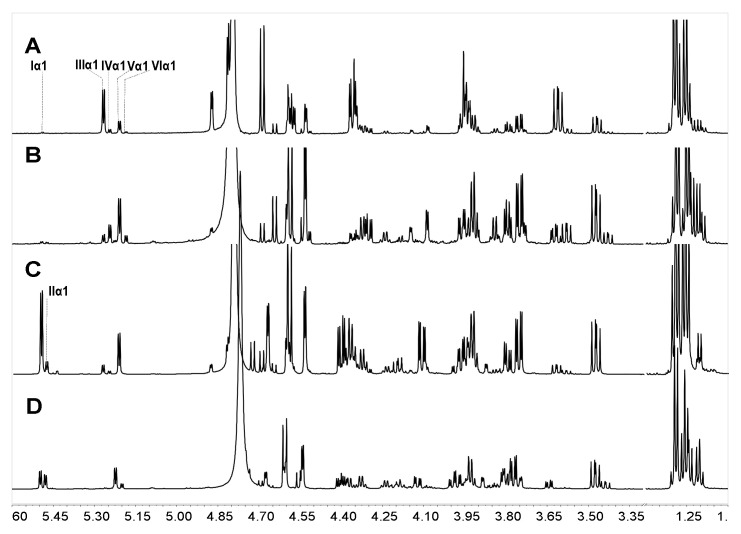
^1^H NMR spectra of Sc19 (**A**), Sc20 (**B**), Sc9 (**C**), and Sc18 (**D**). Sc9 and Sc19 were the side chains obtained at 60 °C for 12 h from SvFG and HfFG, respectively; Sc18 and Sc20 were the side chains obtained at 100 °C for 2 h from SvFG and HfFG, respectively.

**Figure 8 marinedrugs-18-00286-f008:**
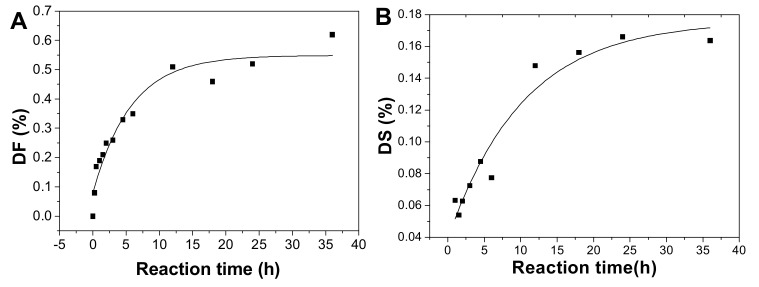
Plot of degree of defucosylation (DF) (**A**), degree of desulfation (DS) in fucose branches (**B**), and degree of hydrolysis (DH) of the backbone glycosidic bonds (**C**) as a function of hydrolysis time.

**Figure 9 marinedrugs-18-00286-f009:**
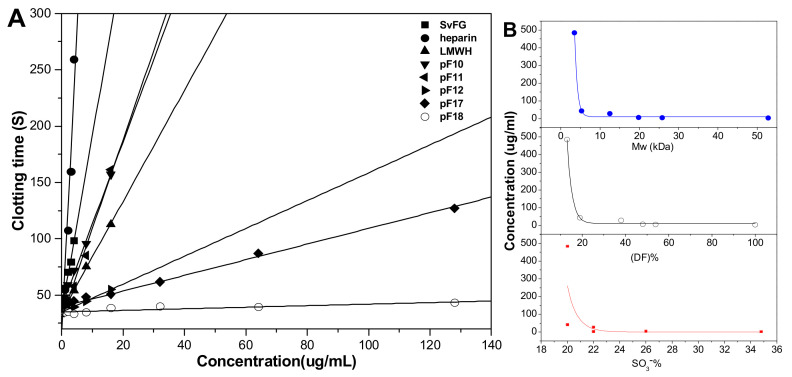
APTT (activated partial thromboplastin time) prolongation activities of heparin, LMWH (low-molecular-weight heparin), native SvFG, and its partial fucosylated derivatives (pF10–12, pF17–18) (**A**); Effects of molecular weight, degree of fucosylation, and –SO_3_^−^ content in pFs on their APTT prolonged activities (**B**).

**Table 1 marinedrugs-18-00286-t001:** Physicochemical properties of mild acid hydrolysis product of SvFG (pF1–18) and HfFG (pF19–20) from different reaction conditions.

Samp.	Treat. (°C × h)	^a^ Yield (%)	^b^ RT (min)	^c^ Mw_t_ (kDa)	^d^ PDI	-SO_3_^−^ (%)	DF (%)	DS (%)	DH (%)
**SvFG**	--	100	14.287	52.77	1.48	ND	0	--	--
**pF1**	60 × 0.25	99.45	14.405	52.45	1.85	36	8	--	--
**pF2**	60 × 0.5	89.32	14.414	53.16	1.64	36	17	--	--
**pF3**	60 × 1.0	77.76	14.465	48.56	1.9	30	19	6.32	-0.17
**pF4**	60 × 1.5	85.73	14.56	46.44	1.81	32	22	5.39	-0.02
**pF5**	60 × 2.0	93.97	14.656	43.44	1.96	28	25	6.28	0.03
**pF6**	60 × 3.0	89.79	14.869	36.52	1.98	28	26	7.25	0.14
**pF7**	60 × 4.0	86.86	15.073	34.79	1.91	28	33	8.75	0.43
**pF8**	60 × 6.0	83.31	15.142	33.43	1.92	26	35	7.75	0.53
**pF9**	60 × 12.0	73.09	15.964	22.93	1.95	22	52	14.78	0.40
**pF10**	60 × 18.0	71.03	15.698	25.85	1.83	22	46	15.62	1.48
**pF11**	60 × 24.0	67.06	16.216	19.83	1.80	26	52	16.61	1.01
**pF12**	60 × 36.0	72.65	16.949	12.54	1.80	22	63	16.36	1.71
**pF13**	50 × 5.0	96.16	14.481	47.72	1.72	22	17	1.00	0.07
**pF14**	60 × 5.0	86.64	14.914	37.06	1.90	26	29	6.17	0.46
**pF15**	70 × 5.0	48.64	15.778	24.94	1.78	20	47	17.49	1.17
**pF16**	80 × 5.0	50.99	16.949	12.91	1.71	24	63	23.50	3.42
**pF17**	100 × 0.5	--	17.120	5.33	1.52	20	81	18.65	10.02
**pF18**	100 × 2.0	--	18.794	3.50	1.47	20	87	34.56	14.60
**pF19**	60 × 12.0	--	15.67	27.66	2.03	26	62	8.91	4.753
**pF20**	100 × 2.0	--	19.035	3.57	1.74	24	91	55.12	12.36

The theoretical Mw of SvFG detected by the Beijing Center for Physical and Chemical Analysis with the polydispersity of 1.22. ^a^ The yield was determined by HPGPC with a standard curve of SvFG. ^b^ Retention time. ^c^ Mw_t_ is the theoretical average molecular weight. ^d^ PDI is the polydispersity.

**Table 2 marinedrugs-18-00286-t002:** ^1^H and ^13^C chemical shifts of sulfated fucose residues obtained by partial acid hydrolysis of SvFG and HfFG.

Resource	Sugar	*J* _H1,H2_ *(Hz)*	Chemical Shift (ppm)				
			H/C-2	H/C-3	H/C-4	H/C-5	H/C-6
***S. variegatus*** **(Sc8)**	α-Fuc2S4S (I)	3.6	4.40/77.92	4.11/68.82	4.67/83.18	4.38/68.02	1.25/18.39
	α-Fuc2S (II)	4.2	4.40/78.13	3.40/69.72	3.88/74.48	4.25/68.70	1.21/17.85
	α-Fuc3,4S (III)	3.6	3.96/68.69	4.58/77.76	4.88/81.48	4.36/68.37	1.25/18.57
	α-Fuc3S (IV)	3.6	3.91/ND^a^	4.53/ND	4.16/ND	4.25/ND	1.22/ND
	α-Fuc4S (V)	4.2	3.81/70.69	3.97/70.84	4.61/83.13	4.33/68.14	1.26/18.50
	β-Fuc2,4S (I’)	7.8	4.20/82.27	3.93/73.28	4.60/82.42	3.94/72.45	1.02/18.45
	β-Fuc3,4S (III’)	7.8	3.62/72.13	4.36/80.62	4.82/80.66	3.96/72.58	1.30/18.60
	β-Fuc3S (IV’)	8.4	3.60/ND	4.31/ND	4.10/72.01	3.85/ND	1.27/ND
	β-Fuc4S (V’)	--	3.50/74.20	3.76/74.52	4.54/82.35	3.93/75.48	1.29/18.56
***H. fuscopunctata*** **(Sc19)**	α-Fuc3,4S (III)	3.6	3.95/68.68	4.59/77.77	4.88/81.46	4.35/68.38	1.26/18.37
	α-Fuc3S (IV)	3.6	3.92/68.52	4.54/80.36	4.15/72.62	4.25/68.57	1.22/18.01
	α-Fuc4S (V)	4.2	3.80/79.71	3.97/70.84	4.61/83.16	4.33/68.15	1.26/18.53
	α-Fuc (VI)	3.6	3.73/ND	3.84/ND	3.80/73.35	ND/ND	1.19/ND
	β-Fuc3,4S (III’)	7.8	3.61/72.16	4.37/80.63	4.82/80.57	3.95/72.48	1.30/18.70
	β-Fuc3S (IV’)	7.8	3.59/72.09	4.31/83.09	4.10/72.09	3.85/72.98	1.25/17.98
	β-Fuc4S (V’)	7.8	3.48/74.21	3.76/74.51	4.54/82.30	3.93/72.49	1.29/18.59
	β-Fuc (VI’)	--	3.43/74.14	3.64/75.36	3.73/73.90	3.80/ND	1.23/ND

ND is not detected.

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
