# Peer review of "Characterization of the Hydrolysis Kinetics of Fucosylated Glycosaminoglycan in Mild Acid and Structures of the Resulting Oligosaccharides"

_marinedrugs, 2020, doi:10.3390/md18060286_

Round 1

Reviewer 1 Report

In this study, the hydrolysis kinetics of fucosylated glycosaminoglycan from S. variegatus in mild acid has been investigated. All the obtained hydrolysed compounds have been thoroughly described by NMR. Authors wanted to define the range of hydrolysis of FG and by this they aim to fully define such tricky polymers. I trust this is a very well conducted research showing a very good rank of polysaccharide analytical chemistry.

I have no remarks since the paper in my opinion is complete and self consistent

Author Response

Thank you very much for your recommendation of our manuscript.

Reviewer 2 Report

This manuscript from Liu et al. characterizes the structure of a fucosylated glycosaminoglycan using proton NMR and studies the anticoagulant potency of various degradation products. Given the complexity associated with the structure of glycosaminoglycans, this work is highly appreciable. 

The authors did good work in structuring the study, however, minor modifications in English language are recommended before publishing. 

The introduction appears to be too long and can be reduced (as deemed necessary by the authors).

Author Response

The authors did good work in structuring the study, however, minor modifications in English language are recommended before publishing. 

We have revised the WHOLE manuscript carefully and try to avoid any grammar or syntax error.

The introduction appears to be too long and can be reduced (as deemed necessary by the authors).

Thanks for the reviewer’s suggestion and the introduction has been reduced in the revised manuscript.

Reviewer 3 Report

The authors (X. Liu, …. J. Zhao) mainly described the detail study on the acid hydrolysis of FCS from two kinds of sea cucumbers. They executed very precisely and accurately the experiments. Sulfation patterns of the original FCS and the ratio of the sulfated fucoses have been changed during hydrolysis. They concluded that the sulfates were hydrolyzed, especially at the specific positions, under acidic and warmed condition. I agree. These results are involved in the study on FCS assignments relating to the biological study. The Reviewer recommends this paper should be accepted to Marine Drugs. Minor revision below should be considered before publication.

  1. Line 130-135, These sentences mentioned on the data of Table 1. Please add the word “Table 1” in the sentence.
  2. Table 1, Optical rotation has no significant meanings for discussion, which should be deleted.
  3. Line 160, Mw of pF1-12 were calculated as 52.45~12.54 kDa. Does the difference (ca.40 kDa) mean the lost weight of fucose branch? Was the CS-like backbone also shortened?
  4. How many Fuc residues linked to the original CS-backbone per disaccharide unit?
  5. Line 190-191, Signal(s) at 5.1 ppm in 1H NMR chart was assigned as H-1(alpha) of GlcA and GalNAc at the reducing terminal. The peak area of this peak is not neglectable. The length of the CS-backbone after hydrolysis would be composed of 30~40 sugar residues (12 kDa). Could you show the peak areas of 5.1 ppm compared to other peaks of CS-like backbone? Alternatively, is it available to assign the 5.1 ppm peaks as internal H-1s which have been epimerized to alpha?
  6. Line 258-259, It should be added that the results were from Figure 7.
  7. The fashion of “D- and L-“ of the sugar residue should not be italic but small capital.

Author Response

Thanks for the reviewer’s suggestion and the reply has been put in the file.
